# Anisakis, Something Is Moving inside the Fish

**DOI:** 10.3390/pathogens11030326

**Published:** 2022-03-07

**Authors:** María Teresa Audicana

**Affiliations:** Department of Allergy and Clinical Immunology, University Hospital of Araba (HUA), OSI ARABA, Basque Country, 01009 Vitoria-Gasteiz, Spain; mariateresa.audicanaberasategui@osakidetza.eus; Tel.: +34-686-84-88-35

**Keywords:** *Anisakis simplex*, allergy, urticaria, anaphylaxis, parasitism, arthritis, mastocytosis, occupational asthma

## Abstract

The first case of human infection by a species of the Anisakidae family was reported more than 60 years ago. Over the last 20 years, *Anisakis* has become a highly studied parasite, not only for its parasitism, but also for its role as an inducer of allergic reactions. Several studies have indicated that the pathological changes occurring within the gastrointestinal tract during infection with *Anisakis simplex* are the combined result of the direct action of the larvae invading the tissue and the complex interaction between the host’s immune system and the parasite. Although the most commonly described pathologies are digestive, urticaria/angioedema and anaphylaxis, occupational asthma and arthritis have been seldom described. This paper is a narrative of the immune-mediated reaction induced by this parasite over the course of the last two decades.

## 1. Introduction

For millennia, humans have consumed fish and products derived from fish. The consumption of raw or undercooked fish carries the risk of transmitting live larvae, especially helminths. Of all the diseases caused by fish helminths, opisthorchiids are the most commonly diagnosed, but its spread is mostly in East and Southeast Asia [1]. On the contrary, cases of human by species of anisakids have been described on all continents. Amongst them, the L3 larval stage of *Anisakis simplex* (*As*) is the most commonly involved in human infections and, less frequently, *Pseudoterranova decipiens* [2]. Only a few *Anisakis physeteris* and *Contracaecum* spp. cases have been described [3,4,5,6].

The first case of human infection by a species of the *Anisakidae* family occurred in the 1960s, when Van Thiel, from the Institute of Tropical Medicine in Leiden, Netherlands, identified patients suffering from severe abdominal pain after ingesting fish [7]. In the original article, the authors erroneously identified it as *Eustoma rotundatum,* and later, in 1962, Van Thiel recognized his error and identified the helminth as *Anisakis* sp. larva [8]. *Anisakis* spp. is a common parasite of marine fish and mammals, and the zoonosis caused in humans was named anisakiasis [2]. Ninety percent of anisakidosis cases are described in Japan, with approximately 20,000 cases per year, reflecting the frequent consumption of raw fish in that country. However, over the last few years, there has been an increase in the number of cases reported in other countries, including Spain, United Kingdom, other European countries, and in the USA [8,9,10,11].

The larvae of *Anisakis simplex* (L3) are visible to the naked eye and have the appearance of a whiteish-pink cylindrical worm, 20 to 30 mm in length. Humans are accidental hosts, interrupting the life cycle of these parasites. The parasite rarely evolves in man to fourth larval stage (L4) and never to adult stage, since the definitive hosts are cetaceans. Among the numerous species of fish and cephalopods that undergo parasitization by anisakids, many of them are commercially important for human consumption. Some of the most important are herring, sardine, anchovy, salmon, haddock, hake, blue whiting, tuna fish, monkfish, turbot, mackerel, horse mackerel and squid, among others. In fact, any marine fish is susceptible to being parasitized by anisakid larvae. These larvae are usually lodged in the visceral (bungling or pelleted) packets of the fish or embedded in the muscles closest to the abdominal cavity and peritoneal cavity (forming flat spirals). Regarding to farmed fish, European and Spanish (APROMAR2012) data indicate the absence of *Anisakis* larvae. However, excluding farmed salmon today, the rest of the fish are not fully guaranteed to be free of *Anisakis* [11].

Ingestion of marine fish and fishery products, containing live *Anisakis simplex* (*As*), causes different clinical conditions, depending on the degree of penetration into the mucosa: luminal or non-invasive form (more frequent in *Pseudoterranova*), and the invasive form that is predominant in the *Anisakis* genus. Furthermore, there are two well-differentiated clinical conditions, depending on the affected segment of the digestive tract: gastric or intestinal. Live *Anisakis* anchors to the wall of the digestive tract and releases enzymes that cause tissue damage, associated with local eosinophilia, that can be followed by eosinophilic granuloma, intestinal perforation and/or severe allergic reactions. [12,13]. In 1990, Japanese authors were first to suggest the involvement of these parasites in allergic reactions that occur in the context of acute parasitism, with urticarial and anaphylactoid syndromes after eating raw fish [14]. After the report of the first case of IgE-mediated anaphylaxis, confirmed by in vivo and in vitro tests with *As*, our group reported several patients with immediate hypersensitivity developed after the ingestion of a parasitized fish [12,13,15].

In Spain, there have been many reports of cases since 1995, probably since the Spanish allergists were mindful of this new pathogen. These reports described cases with clinical symptoms that cover the entire spectrum of the allergic response, from isolated skin symptoms (urticaria and angioedema) to the most severe ones, such as anaphylaxis. *Anisakis* is considered to be the main food allergy factor associated with urticaria and angioedema in adults, especially in non-atopic-middle-aged patients in the north of Spain. This parasite is responsible for 8% of acute urticarial/angioedema cases and 11% of anaphylactic episodes [13]. Data show that in the adult population, allergy to this parasite yields similar appearance to other common allergens, such as shellfish or nuts. Fortunately, most cases are non-life-threatening hives. However, it is noteworthy that most patients (65%) required treatment in the emergency room and some patients were even admitted to Intensive Care Units [14,15,16,17,18,19]. Nowadays, skin prick tests are a common practice, allowing results in 15–20 min when testing for *Anisakis simplex* hypersensitivity. These allergens are included in the standard sets for the investigation of food allergies, anaphylaxis and even drug allergies when drugs are ingested close in time with fish.

The main allergic and immunological pathologies associated with this parasite are listed below.

## 2. Allergic Symptoms

The clinical history of allergy to *As* is not as clear as it might be in other types of food allergies. Most patients do not associate allergic symptoms with eating fish because they have previously tolerated fish for decades, often between episodes. Moreover, this parasite has been considered, in recent years, as a hidden allergen [18]. The acute allergic reaction is usually triggered between 15 and 30 min, up to 2–6 h after consuming fish. It more frequently occurs in adult subjects, between 40 and 70 years. This reaction can range from a simple episode that simulates gastroenteritis to an anaphylactic shock, requiring treatment in the intensive care unit [19]. Some *As* allergens are highly resistant to heat and freezing and, therefore, despite these measures, larval death cannot be ensured, thus, protection against allergic episodes is not guaranteed.

The reported percentages of cutaneous, respiratory and hypotension/syncope manifestations are very comparable to other series of anaphylaxis (100%, 39% and 23%, respectively). It is important to highlight that skin symptoms are the most frequent (almost 100% of patients), and digestive symptoms are also observed (74%) [15,16,17,18,19].

It has been shown that the levels of specific and total IgE increase considerably in the following days after the parasitic infection and remain elevated for months or even years [12,13]. This fact led to the commercialization of specific IgE to *As*, initially to detect the parasite infection and, more recently, it has been used in allergy diagnosis. Type I hypersensitivity or allergic response is usually diagnosed by in vivo test (prick tests) and a subsequent confirmation with in vitro tests (specific IgE, histamine release and/or basophil activation test (BAT)). In the case of *As*, the skin prick test was used for the first time in 1995, and since then, its use has become widespread among allergists, as a valuable screening test in cases of urticaria and anaphylaxis, following the guidelines of the European and American and WAO Academies of Allergy [15]. In recent decades, allergen-induced BAT detection by flow cytometry has been proposed as a useful diagnostic technique for food and drug allergy [20]. This test uses whole blood samples and live basophils, detecting activation-associated membrane markers (CD63) after the *As* extract binds to the cell surface IgE. BAT results showed very high intensities (about 90%) for *Anisakis* extracts in studies of symptomatic versus asymptomatic patients [13].

The risk of *Anisakis*-associated hypersensitivity from ingestion of properly cooked and frozen fish has been controversial. It is the dilemma of the living or dead parasite and its ability to induce allergic response. However, anaphylaxis has been described by prick testing with *Anisakis* extracts, which confirms the evidence that purified *As* allergens are potent enough to cause severe allergy in some patients. In addition, it has been shown that there are trace amounts of *As* allergens present in the fish in the areas of the muscle adjacent to where the larva was present [13]. These reports support the hypothesis that parasitic antigens may be present in the edible muscle of fish, eliciting an allergic response without the need for concomitant infection [21].

Some authors have suggested that *Pseudoterranova decipiens* larvae, especially those found in the United States, are less invasive and less pathogenic than *As* larvae [22]. It has been suggested that *Contracaecum* sp could give rise to allergic reactions after sensitization in animals, due to the presence of allergens shared with *As*. Recently, this IgE-cross-reactivity has been confirmed in some patients by skin tests and in vitro tests [23].

## 3. Sensitization without Allergy Symptoms

Several available serodiagnostic tests for *Anisakis* are based on diagnostic methods that cause cross-reactivity due to the use of unfractionated or partially purified antigens and, thus, show poor specificity, due to cross-reactivity with antigens from many other parasites [13]. Specific IgE was developed to avoid this problem and yet it is not uncommon to detect low levels in asymptomatic subjects, and even in 25% of healthy controls [13]. Possible explanations could be the presence of a ‘tropomyosin”, as a panallergen, present in crustaceans, insects and mites, or cross-reactivity due to other parasites, carbohydrates, phosphoryl choline, glycans or biotinyl-enzymes that can stimulate the production of IgE in some patients.

Not only IgE antibodies are detectable in healthy individuals, but high levels of IgG1 antibodies, reactive with biotinyl-enzymes (BE) from nematodes that are also detected by capture-ELISA, using streptavidin as a specific ligand. Biotinyl-enzymes are well-conserved molecules present in helminths, as well as in other animals, bacteria and plants [24]. Carbohydrates may be present in parasite glycoproteins and a deglycosylated antigenic fraction, named UA3R, improves the specificity of diagnosis of parasite infection versus allergy [25].

Although low levels of *As*-specific-IgE cannot be considered allergy specific, high levels do correlate with allergic symptoms [12,13,26]. To improve the specificity of serological diagnosis, it would be interesting to have a more concrete test, perhaps based on a cocktail of recombinant proteins.

## 4. Occupational Asthma and Dermatitis

Cases of occupational allergy have been reported in Spain, Italy and South Africa [27,28,29,30,31,32]. Rhinoconjunctivitis and occupational asthma caused by *As* has been described in fishmongers, fishermen, processing factories, farmers exposed to fishmeal and an isolated case involving a housewife. A recent review of occupational allergens includes this parasite as an infrequent case for reaction. These authors suggest that sensitization by inhalation is less frequent or remains undiagnosed. Supporting this hypothesis of underdiagnosed pathology, a South African study shows that 50% of fishermen are sensitized to *As* [32].

Similarly, protein contact dermatitis has been described by this parasite in restaurant workers and other people working in contact with fish. There has also been a confirmed case involving a housewife with type I (IgE-mediated) and type IV (delayed) response [33]. Therefore, housewives could join the population at risk of sensitization, as other professionals described above. Thus, two types of professions are associated with allergy to *Anisakis*, those who manipulate fish (captures, processing, sale, restoration, housewife) and the ones who work in contact with food destined for animals, which is commonly made of fish flour.

## 5. Mastocytic Activation Syndrome and Mastocytosis

Mast cells are the key cells in the allergic response as they release the inflammatory mediators responsible for allergic symptoms. A prospective study on mastocytosis in adults and children showed that the frequency of sensitization to *As* was 26% in adults but only 13.6% of the patients presented associated symptomatology. In children, however, IgE test with *Anisakis* was negative in all cases [34].

## 6. Rheumatological Symptoms

A few rheumatological symptoms were described within the first cases of *Anisakis* infection and posteriorly, those associated to allergic reactions. These patients presented simultaneous reactive arthritis (with consequent positive acute phase reactants), along with cutaneous manifestations [35]. There is scarce literature linking urticaria/angioedema with arthralgia and only a few rare cases of anaphylaxis were reported.

## 7. Conclusions

To summarize, much progress has been made in understanding the pathology induced by *Anisakis simplex*, but there are still several issues to be resolved. In the coming years, a significant advance is expected, both in the evolution of rapid diagnostic techniques to more efficiently identify patients at risk, and in detecting parasites or their proteins in fish or dishes made with seafood.

At present, health care professionals must be aware of the importance and dangers of zoonoses and, in particular, parasitic diseases, which, unfortunately, for the last half a century, have been considered of low relevance in the Western world.

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
