# Peer review of "Anisakis, Something Is Moving inside the Fish"

_pathogens, 2022, doi:10.3390/pathogens11030326_

Round 1
Reviewer 1 Report
The review „Anisakis, something is moving inside the fish” describes Anisakis sp. as a cousative agent of allergic disease dangerous for human.
The article is well written, it reads well and provide information interesiting for readers from the outside of medicine and parasitology.
I have only one suggestion for the authors: Whether authors could include in the manusript section about diagnostic methods for anisakiasis?
Author Response
The author appreciates the reviewers' suggestions and apologizes for the delay due to illness due to COVID-19.
Comments From Reviewer 1
- The review „Anisakis, something is moving inside the fish” describes Anisakis as a cousative agent of allergic disease dangerous for human.
- The article is well written, it reads well and provide information interesiting for readers from the outside of medicine and parasitology.
- I have only one suggestion for the authors: Whether authors could include in the manusript section about diagnostic methods for anisakiasis?
Question |
Response |
· I have only one suggestion for the authors: Whether authors could include in the manusript section about diagnostic methods for anisakiasis?
|
· This mini-review is not based on anisakiasis but on the contribution from the point of view of allergy in the response to this parasite. · Regarding the expansion of the bibliographic references in relation to suggested diagnostic techniques, the reference number 11, specifically reviews these techniques. |
Reviewer 2 Report
This review presented by Audicana clearly resume the key topics of anisakis allergy and provide to reader an expert opinion. Allergic anisakiasis is an emergent public health issue, especially in high-risk countries where high prevalence in fish have been reported. However, some clinical aspects are still neglected and epidemiological data should be increase. In light of these consideration and after a deep examination, I recommend the publication of this manuscript with only a few minor corrections. Please find more specific comments below.
The numbering of the bibliographic references must be inserted in square bracket, please, correct this error throughout the manuscript
Line 31-33 please add a bibliographic reference
Line 38 please, change adulthood with do not develop to adult stage
Line 42-43 please delete this sentence or add a bibliographic reference
Line 59-61 Which are these reports? In my opinion, for the reader it is important to have the bibliographic references in the text. In conclusions, this paper is presented as a review, therefore I strongly suggest to increase the bibliographic references throughout the manuscript.
Check the reference list carefully again from the beginning. Reference lists are frequently hotbeds of errors. The format adopted for the manuscript is wrong, please correct the references according to mdpi format.
Author Response
The author appreciates the reviewers' suggestions and apologizes for the delay due to illness due to COVID-19.
Comments From Reviewer 2
This review presented by Audicana clearly resume the key topics of anisakis allergy and provide to reader an expert opinion. Allergic anisakiasis is an emergent public health issue, especially in high-risk countries where high prevalence in fish have been reported. However, some clinical aspects are still neglected and epidemiological data should be increase. In light of these consideration and after a deep examination, I recommend the publication of this manuscript with only a few minor corrections. Please find more specific comments below.
The numbering of the bibliographic references must be inserted in square bracket, please, correct this error throughout the manuscript
Line 31-33 please add a bibliographic reference
Line 38 please, change adulthood with do not develop to adult stage
Line 42-43 please delete this sentence or add a bibliographic reference
Line 59-61 Which are these reports? In my opinion, for the reader it is important to have the bibliographic references in the text. In conclusions, this paper is presented as a review, therefore I strongly suggest to increase the bibliographic references throughout the manuscript.
Check the reference list carefully again from the beginning. Reference lists are frequently hotbeds of errors. The format adopted for the manuscript is wrong, please correct the references according to mdpi format.
Question |
Response |
· The numbering of the bibliographic references must be inserted in square bracket, please, correct this error throughout the manuscript
|
· The numbering of the bibliographic references are now inserted in square bracket and mdpi format. |
· Line 31-33 please add a bibliographic reference
|
· Sentences have been changed by indication of the publisher |
· Line 38 please, change adulthood with do not develop to adult stage |
· Done change: do not develop to adult stage |
· Line 42-43 please delete this sentence or add a bibliographic reference |
· Sentences have been changed by indication of the publisher |
· Line 59-61 Which are these reports? In my opinion, for the reader it is important to have the bibliographic references in the text. In conclusions, this paper is presented as a review, therefore I strongly suggest to increase the bibliographic references throughout the manuscript. |
· Sentences have been changed by indication of the publisher. 1. The aim of this minireview is to focus on the novelty of the allergy point of view. 2. A new reference was added: Audicana, M.T.; del Pozo,M.D.; Iglesias, R.; Ubeira, F.M. Anisakis simplex and Pseudoterranova decipiens. In: International handbook of foodborne pathogens. New York: Marcel Dekker, 2003, pp 613-636. · The author appreciates the suggestions but considers that he has already made a revision in reference 11 on more general aspects. · Sentence in line 83 was changed by “ The main allergic and immunological pathologies associated with this parasite are listed below”. |
· Check the reference list carefully again from the beginning. Reference lists are frequently hotbeds of errors. The format adopted for the manuscript is wrong, please correct the references according to mdpi format. |
· References are checked and changed according to mdpi format. |